# *Borrelia burgdorferi* Migration Assays for Evaluation of Chemoattractants in Tick Saliva

**DOI:** 10.3390/pathogens11050530

**Published:** 2022-05-01

**Authors:** Mary B. Jacobs, Britton J. Grasperge, Lara A. Doyle-Meyers, Monica E. Embers

**Affiliations:** 1Division of Immunology, Tulane National Primate Research Center, Tulane University Health Sciences, Covington, LA 70433, USA; mb_jacobs@hotmail.com; 2Department of Pathobiological Sciences, School of Veterinary Medicine, Louisiana State University, Baton Rouge, LA 70802, USA; bjgdvm@gmail.com; 3Division of Veterinary Medicine, Tulane National Primate Research Center, Tulane University Health Sciences, Covington, LA 70433, USA; ldoyle@tulane.edu

**Keywords:** *Borrelia burgdorferi*, *Ixodes* tick, saliva, vector, migration, chemoattractant

## Abstract

Uptake of the Lyme disease spirochete by its tick vector requires not only chemical signals present in the tick’s saliva but a responsive phenotype by the *Borrelia burgdorferi* living in the mammalian host. This is the principle behind xenodiagnosis, wherein pathogen is detected by vector acquisition. To study migration of *B. burgdorferi* toward *Ixodes scapularis* tick saliva, with the goal of identifying chemoattractant molecules, we tested multiple assays and compared migration of host-adapted spirochetes to those cultured in vitro. We tested mammalian host-adapted spirochetes, along with those grown in culture at 34 °C, for their relative attraction to tick saliva or the nutrient N-acetyl-D-glucosamine (D-GlcNAc) and its dimer chitobiose using two different experimental designs. The host-adapted *B. burgdorferi* showed greater preference for tick saliva over the nutrients, whereas the cultured incubator-grown *B. burgdorferi* displayed no significant attraction to saliva versus a significant response to the nutrients. Our results not only describe a validated migration assay for studies of the Lyme disease agent, but provide a further understanding of how growth conditions and phenotype of *B. burgdorferi* are related to vector acquisition.

## 1. Introduction

Lyme borreliosis is a multistage disease resulting in a progression of clinical manifestations from early to late phase infection. Signs of early phase infection, including erythema migrans and flu-like symptoms, begin to present after *Borrelia burgdorferi* (Bb) spirochetes are injected into the host’s skin through the bite of an infected *Ixodes scapularis* tick [1]. As dissemination of spirochetes proceeds, the host may experience arthritis, carditis and detrimental effects to the central and peripheral nervous systems [2,3,4]. These advanced symptoms are indicative of a late phase infection and can persist for many months to years [5,6,7]. Although Lyme disease has traditionally been thought to be easily treatable with common antibiotics, many patients continue to suffer with debilitating non-specific symptoms such as joint pain and cognitive impairment [5]. It remains unclear if the chronic manifestations are due to persistence of viable bacteria within these tissues or chronic immune stimulation by bacterial proteins or genetic material [8]. Nonetheless, the incursion of *Borrelia burgdorferi* spirochetes into immune-privileged sites is a known immune evasion strategy that aids in their survival and persistence.

With this disease and others, one of the most sensitive methods to detect residual infection is xenodiagnosis, wherein the spirochetes lingering in the host are recruited to a naïve tick feeding on the host, and acquired [9,10]. Interestingly, however, the molecular mechanisms that govern this communication between spirochete and tick are largely unknown. Identification of the chemoattractants for *B. burgdorferi* in tick saliva would not only fill a distinct vector biology void for basic science but add tremendous potential to applied science of human *B. burgdorferi* infection. A potent tick salivary chemoattractant would prove a safe measure to enhance Lyme disease diagnosis and therapy by drawing the spirochetes from their shelters.

The vertebrate host and *Ixodes* tick offer greatly diverse environmental conditions to which the Borrelia spirochetes, as obligate symbionts, must differentially adapt to in order to successfully maintain their natural enzootic cycle [11,12]. Part of this cycle, which is crucial to Borrelia survival, includes the motility and chemotaxis responses of the spirochete. These responses are controlled by genes which are modified by environmental cues [13]. Gene expression of in vitro cultured spirochetes can be altered by changes in pH, oxygen, carbon dioxide, various nutrients, and temperature [14]. Additionally, host-adapted cultivation in a dialysis membrane chamber (DMC) implanted into a rat peritoneal cavity results in gene expression changes which are distinct from in vitro cultured spirochetes. Akins et al. demonstrated that spirochetes grown in temperature elevated to mimic that of a mammalian host yielded protein profiles that were different than those that were DMC-cultivated [15]. For this study, we examined the ability of *B. burgdorferi* that were either cultured in vitro or host-adapted to migrate towards tick saliva and carbohydrate sources.

We hypothesized that host-adapted Borrelia may be more readily attracted to the tick saliva than those cultivated in vitro. We also sought to optimize an assay to measure this chemoattractive migration reliably so that specific molecules needed to recruit spirochetes to ticks could be identified.

## 2. Results

The migration of Borrelia towards various chemoattractants was tested using three different assays. Chemoattractants utilized tick saliva as the test with chitobiose and D-GlcNAc as the positive controls. A depiction of the various migration assays is shown in Figure 1.

### 2.1. Assessment of Motility Using the U-Tube

Through a variety of modifications described, we maintained agarose at 0.1% concentration in keeping with reporting by Shih et al. of optimal spirochete migration [16]. Due to lack of any positive migration results, we proceeded to attempt variations in quantity and type of matrix, concentration of cells, chemoattractants and incubation conditions.

Initially, we were concerned about possible interference of carbohydrates in BSK-II which might themselves act as attractants and thereby delay migration of spirochetes through to the opposing chemoattractant side of the U-tube. Use of PBS as a diluent with agarose or chitobiose and in vitro-grown cells for a 24 h period was attempted three times. This environment was not found to be conducive to the movement of spirochetes. Modification of BSK-II medium for diluent and matrix base to lower its sugar content, while adequate for spirochete health at least in the short term, did not lead to any observable cell migration. This modification was used with both in vitro and host-adapted cells against chitobiose, D-GlcNAc or tick saliva.

Finally, we chose to test gelatin as a matrix for motility, replacing the semi-solid agarose medium in this experimental design. Five and 10% gelatin in PBS was tested at both ambient temperature or in a 34 °C trigas incubator using in vitro-cultured or host-adapted spirochetes over the course of up to 7 days. Gelatin is viscous at 34 °C while remaining solid at 23 °C. In addition, a matrix of 4% gelatin in modified low sugar BSK-II was used to test attraction of host-adapted cells to chitobiose at 23 °C over a 48 h period. In all cases of gelatin matrix use, no cells were found to migrate to the chemoattractant. The results are summarized in Appendix A.

### 2.2. The Gelatin Well Assay of Spirochete Migration

In vitro-cultured cells were attracted to chitobiose and D-GlcNAc by as much as a 13:1 ratio vs. control. In similarly structured experiments, these cells showed no preference for tick saliva (5–9%) over PBS background. Ultimately, after running side-by-side comparisons varying experimental specifications, we found optimal conditions for host-adapted cells included a gelatin matrix of 4%, well diameter of 4 mm, well depth of 3 mm and well spacing of 1 mm. The diameter of 4 mm was chosen over 2 mm for the purpose of volume retention following overnight incubation. In contrast to data acquired from in vitro grown cells, saliva was successfully used as a chemoattractant for host-adapted cells at both 5 and 10% concentrations, resulting in a ratio of as much as 17:1 over background (Table 1). D-GlcNAc and chitobiose in this scenario showed minimal or no chemoattractant ability for host-adapted cells.

### 2.3. The Capillary Tube Migration Assay

Although the thickening agent methylcellulose has been added to motility buffer by some investigators, as Shi et al. [17] found no effect on motility, we also omitted its addition. Further, since we wanted to test the buffer with gelatin as a constituent, it might have interfered by creating an even more viscous environment than intended. Similar phenomena were seen in the case of gelatin well assays described above regarding differing chemotactic response. N-acetyl-D-glucosamine at 100 mM was established as a strong attractant for *Borrelia burgdorferi* [18]. This was confirmed in our assay using in vitro grown cells in both motility buffer alone at ambient temperature and gelatin-infused motility buffer assays at 34 °C. Again, tick saliva failed to elicit a chemotactic response by in vitro cells in either matrix environment. Conversely, when host-adapted cells were tested at room temperature in motility buffer, saliva induced a response of approximately 7.5:1 over buffer control. D-GlcNAc, although positive, resulted in a more modest response (Table 2). Given that this was a reproducible assay for evaluating chemoattraction, we describe in detail the methods which were used (Appendix B).

## 3. Discussion

When an infected *Ixodes* tick feeds on an uninfected mammalian host, the pathogenic *B. burgdorferi* spirochetes migrate from the tick midgut to salivary glands and are introduced to the host through the tick saliva. The spirochetes then disseminate throughout the host and can cause disease. In order for the enzootic cycle to continue, uninfected ticks must also be able to acquire *B. burgdorferi* by feeding on infected mammalian hosts. However, other than during the initial inoculation and dissemination of the spirochetes, *B. burgdorferi* is not found in high numbers in the blood. The tick must signal to spirochetes located in tissues such as skin, joints, heart, brain, and bladder to migrate into the blood and to the tick bite site in order to be taken up by the tick. Recently, the Ixodid tick salivary gland protein, Salp12 was shown to be involved in the chemoattraction of spirochetes in the mammalian host by the ticks. Knockdown of Salp12 in ticks or passive immunization of mice with anti-Salp 12 antibodies reduced tick acquisition of *B. burgdorferi* spirochetes [19]. More recently, a number of chemoattractant molecules were successfully tested using a transwell assay [20].

Culturing Borrelia from a persistently-infected host is rarely productive due to slow growth of the pathogen, dissemination out of the bloodstream and residence within immune-privileged sites. Probably the most sensitive way to detect this persistent pathogen is xenodiagnosis. Here, an uninfected vector (in this case, *Ixodes* ticks) feeds on a host and acquires the pathogen, which may otherwise be undetectable. We have successfully used xenodiagnosis for detection of persistent *B. burgdorferi* spirochetes in Lyme disease research. A clear need in this line of research is the lack of a safe and minimally invasive procedure to assess human patients for the presence of persistent, viable Lyme disease spirochetes. To fill this void, one group has begun experimenting with xenodiagnoses in human Lyme disease by feeding naïve *Ixodes scapularis* on the patients followed by examination of the ticks for Borrelia [21]. While this technique is promising, the extensive feeding period required is objectionable to many people, and the inconsistent nature of tick feeding and salivation creates an intractable experimental environment. While murine skin may harbor appreciable levels of spirochetes for ready acquisition by ticks, incidental hosts have a lower abundance of *B. burgdorferi* in skin and blood following initial infection. We surmise that the chemoattraction must be very potent for xenodiagnosis in incidental hosts to be successful.

Our initial interest in examining chemoattraction of Borrelia spirochetes to tick saliva arose due to the importance of the xenodiagnosis technique in our own research to detect persistent pathogens after antibiotic treatment [9]. Initially, we wanted to look at chemotaxis of *B. burgdorferi* to tick saliva in comparison with that of know well-researched carbohydrate attractants, realizing that there are different circumstances under which spirochetes are attracted to carbohydrates versus tick saliva. Growth in a laboratory requires specific metabolic constituents, GlcNAc among them. A variety of carbohydrates would be available to the spirochete from either the vertebrate host as a component of the extracellular matrix (ECM) or arthropod host as chitin. These sugars are utilized by *B. burgdorferi* both as an energy source and as a required structural component for cell wall synthesis. Ultimately, the overall goal is to use this motility assay to identify chemoattractants in tick saliva, which could be used in humans to improve diagnosis (via skin patch or subdermal injection/implant), given the small numbers of spirochetes present in skin biopsies, especially as the infection increases in duration [22].

A caveat to this study is the use of saliva from partially-fed adult female ticks only. The molecules secreted by ticks likely change over the feeding period, which usually lasts about 72 h [23]. Also, spirochetes are acquired by larval and nymphal ticks from the infected host; their salivary composition likely differs from that of adult ticks. Currently, the only feasible method to acquire *Ixodes scapularis* saliva in enough quantity to test is to milk it from partially-fed adult ticks. We did attempt to collect saliva from nymphal ticks but the amount was so minimal that it dissipated from the capillary tube (data not shown). An alternative is to use extracts from dissected salivary glands. However, we surmise that the feeding process induces secretion of molecules not found in salivary glands at a single time point. In addition, growth in dialysis membrane chambers was used to host-adapt the spirochetes, which may not wholly mimic the host environment, given the restriction of the membrane to small molecules. While we make an effort to add the host-adapted spirochetes directly to the assay from incubation within the rat host, we did not confirm the host-adapted phenotype.

We used several methods to assess the migration of Borrelia spirochetes toward chemoattractant molecules. Ultimately, we wanted to devise an assay to test *B. burgdorferi* migration toward tick saliva and to that end, compared in vitro vs. host-adapted *B. burgdorferi.* The chemoattractants studied, especially tick saliva, had different capacities to elicit the migratory response of *B. burgdorferi* cells depending on whether they were grown in vitro or were host-adapted. In summary, our major findings were that (1) host-adapted *B. burgdorferi* more readily migrated towards tick saliva than did in vitro-cultured cells; and (2) that the capillary tube migration assay provided reproducible results, and thus provided a tractable method for future studies. This assay system will be used for the identification of the molecules in *I. scapularis* saliva responsible for chemoattraction of *B. burgdorferi*, aiding research into the elusive etiology of chronic Lyme disease.

## 4. Materials and Methods

Critical to the success of identifying chemoattractants in tick saliva is a reliable assay for directional motility of *B. burgdorferi.* Multiple platforms have been devised to test directional motility of *B. burgdorferi* toward positive control molecules—either N-acetyl-D-glucosamine (D-GlcNAc) or chitobiose. Among these include the U-tube, the gelatin well assay, and the capillary tube assay. Modifications and troubleshooting were applied to each of these to identify the optimal assay giving reproducible results.

### 4.1. Ethics Statement

Practices in the housing and care of rabbits and rats conformed to the regulations and standards of the Public Health Service Policy on Humane Care and Use of Laboratory Animals, and the Guide for the Care and Use of Laboratory Animals. The Tulane National Primate Research Center (TNPRC) and the LSU School of Veterinary Medicine are fully accredited by the Association for the Assessment and Accreditation of Laboratory Animal Care-International. Institutional Animal Care and Use Committees at Tulane and LSU approved all animal-related protocols, including the tick feeding on rabbits and DMC implantation surgery in rats. At Tulane, the rat surgeries were performed by a laboratory animal veterinarian, using all appropriate anesthetics and analgesics.

### 4.2. Bacterial Strain, Media and Growth Conditions

Low passage *B. burgdorferi* wild-type strain B31 5A19 was grown at 34 °C in a trigas incubator (5% CO_2_, 3% air, and the remainder N_2_) in BSK-II supplemented with 6% rabbit serum (Pel-Freez Biologicals, Rogers, AR, USA) and antibiotics as described [24]. Host-adapted *B. burgdorferi* were prepared by use of dialysis membrane chambers in rats as described by Akins et al. [15,25] whereby spirochetes were allowed to transition in the animal host over the course of 10 days. Surgeries were performed to implant the dialysis bags into rat peritonea. The incisions were sealed in double layers whereby the abdominal wall (linea alba) was closed with absorbable sutures and the skin layer was closed with glue and staples. Observation was carried out during the entire procedure and post-operatively. Anesthesia was provided by 1–1.5% isoflurane in oxygen by inhalation, and as a pre-emptive analgesia, rats were given sustained-release buprenorphine via subcutaneous injection. Once the dialysis bags were removed from the rat host, the spirochetes were then used on the same day in chemotaxis assays, described below. Immediately after rat euthanasia and collection of samples from the dialysis bags, the spirochetes were counted, concentrated and placed in a pre-assembled chemotaxis assay. In all cases, concentration of cells by centrifugation at 3900× *g* for 6 min at 23 °C was carried out in media from the dialysis chamber and was not supplemented with fresh media.

### 4.3. Chemoattractants and Matrices

N-acetyl-D-glucosamine (D-GlcNAc) was made to a stock of 500 mM and used at a final concentration of 100 mM. N,N′-Diacetylchitobiose (chitobiose) was made to a stock of 50 mM and used at a final concentration of 10 mM. Both were obtained from Sigma (St. Louis, MO, USA). Agarose LE (Lonza, Rockland, ME, USA) was made to a stock of 1% in ultrapure water then autoclaved for sterility. It was melted by microwave for reuse. Type A gelatin from porcine skin (Sigma, St. Louis, MO, USA) was diluted in sterile PBS (Gibco, Waltham, MA, USA) for gelatin well assays or sterile water for capillary tube assays.

### 4.4. Saliva Collection

Tick saliva was collected according to previously published protocols [26]. Briefly, adult female *I. scapularis* ticks obtained from BEI resources were placed on adult New Zealand White Rabbits (Charles River Laboratories) using an approximately 10 cm diameter hard plastic screw top containment device. This was attached by sticking flaps of wound tape onto the device both inside and out and radiating from the bottom opening of the plastic. After the rabbit was shaved, the device was placed over the area with the tape flaps extending out across the body. Multiple strips of wound tape were placed over the radiating flaps to secure the device to the skin of the rabbit. Several long strips of tape were used to wrap entirely around the torso of the rabbit on either side of the containment device. Stretch wrap was then wrapped around all of the taped portions of the device and the torso of the rabbit being cautious to leave slack in the wrapping. The screw top lid was sealed with tape once the ticks were placed on the rabbit, removed during feeding checks, and replaced for any intervals in the feeding cycle. The rabbits were housed individually in cages placed over soapy water for tick security. The ticks were gently removed shortly before drop off (approximately five to seven days after attachment, where drop off is usually between seven and nine days). Each tick was adhered to a glass slide with scotch tape, and 5 µL of 5% pilocarpine solution was pipetted on the dorsal surface (often wicking under the tape). Capillary tubes were pulled over a flame and allowed to cool, then broken with forceps to appropriately sized openings based on the size of the ticks’ mouthparts as viewed under a dissecting microscope. A small ball of modeling clay was placed at the frosted edge of the slide, while the tick was taped to the opposite end with its mouthparts facing the clay. The slides were placed at an angle for the mouthparts to face down in a plastic container, which was then incubated in a dark chamber with high humidity. Saliva collection was monitored for up to four hours. Saliva was expelled into 2 mL cryotubes, and 0.1 volumes of protease inhibitor cocktail were added. Saliva was then stored at −80 °C.

### 4.5. Chemotaxis Assays

#### 4.5.1. U-Tube Assay

The U-tube assay to measure chemotactic migration of *B. burgdorferi* toward salivary gland extract was reported by Shih et al [16]. We followed the procedures described to measure the chemoattractant potential of D-GlcNAc, chitobiose and tick saliva. Due to lack of any positive migration results, we attempted a variety of modifications. Our initial conditions for assay included the use of a U-shaped glass tube (Sigma, St. Louis, MO, USA) filled with 0.3 mL matrix consisting of complete BSK-II medium supplemented with a final concentration of 0.1% agarose. Tubes held an approximate volume of 700 µL, with measurements shown in Figure 1A. To one side of the matrix, 0.1 mL of spirochete culture at 2–3 × 10^7^/mL was added. On the other side, the test inoculum consisting of either PBS as negative control or chemoattractant (100 mM D-GlcNAc, 10 mM chitobiose or 1–5% tick saliva). U-tubes were secured upright and openings were covered with aluminum foil, then placed in a 34 °C trigas incubator for up to five days. At time points every 24 h, a 5 µL volume was removed from within each inoculated side above the medium of each tube and examined at 500× magnification under a dark-field microscope (DIALUX, Leitz Wetzlar, Midland, ON, USA). Variations in matrix constituents were made to try to optimize possible spirochete migration. One modification included addition of 0.1% agarose in PBS alone, foregoing the use of BSK-II to mitigate any possible interference from sugars in the base medium which might act as a weak attractant to spirochetes. In an alternate attempt to dilute the nutrient-rich medium, a second variation included use of a modified BSK-II base medium such that the carbohydrates were reduced from the original concentrations to 50% of the standard amount of pyruvic acid, 10% of the glucose concentration and 50% of the N-acetyl D-glucosamine concentration. This low sugar BSK-II modification was used for the control and chemoattractant diluent. Notably, the same chemoattractant used in many of the assays as found in BSK-II medium, namely N-acetyl glucosamine, was at a low enough concentration in the medium (1.8 mM) that it should not have acted to interfere with the through-movement of spirochetes. Both in vitro and host-adapted *B. burgdorferi* were tested against all three chemoattractants, including 5% tick saliva. In addition, several assays were attempted using either 4%, 5% or 10% gelatin in PBS as a matrix (data not shown).

#### 4.5.2. Gelatin Well Assay

Mousseau et.al [27] described a chemotaxis assay utilizing adjacent agarose gel wells. An early attempt was made using a similar configuration before switching to gelatin as a matrix for cell migration. Gelatin was dissolved in PBS at a concentration of 4 or 5% in a 50 °C water bath, then poured into the well of a 6-well plate to a depth of 2 mm or 3 mm. Gel plates were cured overnight at 4 °C. Prior to use, they were removed to room temperature and wells were made by removing gelatin plug with a 6 or 4 mm AcuPunch Biopsy Punch (Acuderm Inc., Ft. Lauderdale, FL, USA) in direct correspondence to well depth as stated above. Any remaining gelatin in wells was removed with a pipet tip. Wells were separated by a distance of 1 mm (Figure 1B). Cells, chemoattractants or controls were added to wells at a volume of 30 µL for 4 mm diameter well, and 45 µL for a 6 mm diameter well. In vitro-cultured *B. burgdorferi* were added to the center wells at a concentration of 7 × 10^7^/mL. Host-adapted cells were added at 2 × 10^7^/mL (due to availability constraints). Sterile PBS was added to control wells. Chemoattractants tested in this set-up included D-GlcNAc at 100 mM, chitobiose at 10 mM or tick saliva at 5–10%, diluted in sterile PBS. All assays were carried out at 23 °C in a humid environment consisting of wet Kimwipes (Kimtech^®^) surrounding the plates enclosed within a small Styrofoam box. Samples (control vs. chemoattractant) were examined after approximately 24 h. Samples were removed to vials from which ~5 µL was placed on a slide and total number of cells under the entire coverslip (22 mm × 22 mm) were counted under a darkfield microscope at approximately 300×. In the cases where number/field is indicated (Table 2), 20 fields were counted and averaged.

#### 4.5.3. Capillary Tube Assay

The capillary tube assay to measure bacterial chemotaxis was originally described by Adler [28,29]. We followed modifications similar to those tailored for chemotactic studies of *B. burgdorferi* by Shi et al. [17] and Bakker et al. [18]. The spirochetes were grown to mid-log phase then centrifuged at 23 °C for 6 min at 3900× *g* then gently resuspended to 1 × 10^7^/mL in a motility buffer consisting of 150 mM NaCl, 10 mM NaH_2_PO_4_ (pH 7.6), 0.1 mM EDTA, supplemented with 2% bovine serum albumin (Probumin, Millipore, St. Louis, MO, USA). Host-adapted cells were likewise concentrated for use in assays described. The chemotaxis chamber set up was based on the description by Bakker [18] in which a hole was punched through the lid of a 1.8 mL snap-cap microtube (Phenix Research Products/Thomas Scientific, Swedesboro, NJ, USA), Parafilm™ was placed under the closed lid and a capillary tube with chemoattractant was inserted through the hole. Cells diluted in motility buffer were added to the capped tube to a total volume of 0.15 mL. Appropriately filled capillary tubes (BF120-94-10, Sutter Instruments, Novato, CA, USA) containing 20 µL chemoattractant or plain motility buffer (control) were plugged with silicone grease then inserted into the cell-containing microtube (Figure 1C). This set-up was secured horizontally, and incubated at 23 °C on a benchtop for 2 h. Each capillary tube was then removed and carefully wiped off with a Kimwipe to remove excess cells from the outside for final cell count accuracy. The tube contents were removed by pipet using gel-loading pipet tips and collected into individual tubes. Contents of each tube were gently but thoroughly mixed for homogeneity, then 5 µL of volume placed on a slide and counted at 500× magnification under a dark-field microscope. Counts from throughout the slide were taken to find the average number of spirochetes per field. Each condition (chemoattractant) was carried out in triplicate unless otherwise stated. Thus, each number represented 27 counts: 9 field counted per sample, and counting performed in triplicate.

In a second set of experiments, a 6% solution of gelatin in sterile water was added to a final concentration of 3% to the motility buffer for capillary tube assays with gelatin as part of the matrix. For experiments consisting of motility buffer + gelatin as the cell matrix and chemoattractant diluent, two variations were tested using in vitro grown cells. Details are similar as described above except that in one case, the tubes were incubated for 3 h at 34 °C in a trigas incubator. Alternatively, the set up was incubated at room temperature for 27 h. The latter extended time was to account for the fact that the gelatin-containing matrix solidified quickly at room temperature. Contents from these tubes were removed by first gently heating each capillary tube over a hot plate for 30–60 s. Contents from both assay variations were collected, then counted as described.

## Figures and Tables

**Figure 1 pathogens-11-00530-f001:**
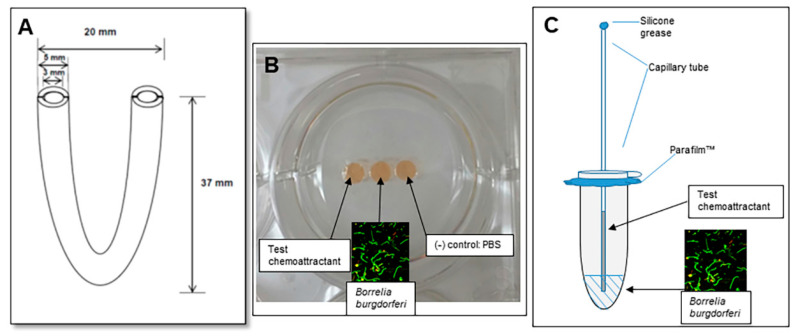
Three different assays for testing migration of *B. burgdorferi* toward chemoattractants. In panel (**A**), a diagram of the U-tube with dimensions. In panel (**B**), an image of the gelatin well assay is shown. While PBS was used in the actual assays, red BSK media was used here for illustration. In Panel (**C**), a depiction of the capillary tube migration assay is provided. The spirochetes in liquid are placed in the bottom of the microfuge tube and the chemoattraction test sample is placed in the capillary tube.

**Table 1 pathogens-11-00530-t001:** Summary of migration results using the gelatin well assay.

Matrix: Gelatin (%)	Well Diameter (mm)	Well Depth (mm)	Distance between Wells (mm)	Cells	[Cell] (×10^7^/mL)	Lateral Well Inocula	[CA]	Volume per Well (µL)	Incubation Time (Hours)	Final Cell Count *	Ratio (CA:Control)
3	6	2	4	in vitro Bb	8	chitobiose	10 mM	35	21	0.45/field	
						PBS				0	
3	6	2	4	in vitro Bb	8	chitobiose	10 mM	35	21	0.9/field	(-)
						BSK-II				1/field	
5	6	2	2	in vitro Bb	8	chitobiose	10 mM	35	21	1.5/field	
						PBS				0	
5	6	2	2	in vitro Bb	8	chitobiose	10 mM	35	21	14/cs	7:1
						BSK-II				2/cs	
5	6	2	2	in vitro Bb	8	tick saliva	5%	40	24	1/cs	1:1
						PBS				1/cs	
5	6	2	2	in vitro Bb	4.6	chitobiose	10 mM	40	21	1/cs	1:1
						PBS				1/cs	
5	6	2	2	in vitro Bb	4.6	tick saliva	5%	40	21	3/cs	(-)
						PBS				8/cs	
5	6	2	1	in vitro Bb	7	chitobiose	10 mM	40	26	20/cs	10:1
						PBS				2/cs	
5	6	2	1	in vitro Bb	7	tick saliva	9%	45	27	2/cs	1:1
						PBS				2/cs	
5	6	2	1	in vitro Bb	7	chitobiose	10 mM	45	27	24/cs	3.5:1
						PBS				7/cs	
5	6	2	1	in vitro Bb	7	chitobiose	10 mM	45	27	25/cs	13:1
						PBS				2/cs	
5	6	2	1	in vitro Bb	7	D-GlcNAc	100 mM	45	26	64/cs	4:1
						PBS				16/cs	
5	6	2	1	in vitro Bb	7	chitobiose	10 mM	45	28	62/cs	1.6:1
						PBS				40/cs	
5	6	4	1	in vitro Bb	7	chitobiose	10 mM	46	27	27/cs	2.5:1
						PBS				11/cs	
4	6	2	1	in vitro Bb	7	chitobiose	10 mM	40	23	37/cs	5.3:1
4	4	3	1	in vitro Bb	7	D-GlcNAc	100 mM	30	22	57/cs	3.2:1
						PBS				18/cs	
5	4	3	1	in vitro Bb	7	D-GlcNAc	100 mM	30	22	21/cs	7:1
						PBS				3/cs	
4	4	3	1	Host-adapted Bb	2	chitobiose	10 mM	30	22	3/cs	
						PBS				0	
4	4	3	1	Host-adapted Bb	2	chitobiose	10 mM	30	22	3/cs	
						PBS				1/cs	
4	4	3	1	Host-adapted Bb	2	tick saliva	5%	30	27	15/cs	7.5:1
						PBS				2/cs	
4	4	3	1	Host-adapted Bb	2	tick saliva	10%	30	28	17/cs	17:1
						PBS				1/cs	
4	4	3	1	Host-adapted Bb	2	D-GlcNAc	100 mM	30	26	3/cs	1:1

* cs, entire area under the coverslip; CA, chemoattractant

**Table 2 pathogens-11-00530-t002:** Capillary tube migration results.

Matrix	Cells	Cap Tube Contents: CA or Control	[CA]	Cap Tube Volume (µL)	No. Replicates	Incubation Environment	Incubation Time (Hours)	AVG. Cell Count per Field	Ratio (CA:Control)
Motility	Host-adapted	D-GlcNAc	100 mM	20	1			1.4	3.5:1
Buffer	Bb	tick saliva	10%	20	3	RT	2	3.1	7.8:1
		MB buffer		20	3			0.4	
Motility	Host-adapted	D-GlcNAc	100 mM	20	1			4.8	5.3:1
Buffer	Bb	tick saliva	10%	20	4	RT	2	6.6	7.3:1
		MB buffer		20	3			0.9	
Motility	in vitro Bb	D-GlcNAc	100 mM	20	3	RT	2	20.8	14.9:1
Buffer		MB buffer		20	3			1.4	
Motility	in vitro Bb	D-GlcNAc	100 mM	20	3			22.3	9.7:1
Buffer		tick saliva	10%	20	3	RT	2	3.9	1.7:1
		MB buffer		20	3			2.3	
Motility	in vitro Bb	D-GlcNAc	100 mM	20	3	RT	2	13	26:1
Buffer		MB buffer		20	3			0.5	
Motility Buffer+	in vitro Bb	D-GlcNAc	100 mM	20	3	34 °C trigas	3	4.9	5.5:1
3% gelatin		MB/gelatin		20	3			0.9	
Motility Buffer+	in vitro Bb	D-GlcNAc	100 mM	20	3	34 °C trigas	3	9.4	5.9:1
3% gelatin		tick saliva	10%	20	3			2.1	1.3:1
		MB/gelatin		20	3			1.6	
Motility Buffer+	in vitro Bb	D-GlcNAc	100 mM	20	3	34 °C trigas	3	7.4	5.7:1
3% gelatin		MB/gelatin		20	3			1.3	
Motility Buffer+	in vitro Bb	D-GlcNAc	100 mM	20	3		27	1.5	1.5:1
3% gelatin		tick saliva	10%	20	3	RT		1.1	1.1:1
		MB/gelatin		20	3			1	

Bb = Borrelia burgdorferi; CA = chemoattractant.

## Data Availability

The article contains all datasets in which significant results were observed. Supplemental Table 1 includes results for which migration was not observed.

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
