# Peer review of "Borrelia burgdorferi Migration Assays for Evaluation of Chemoattractants in Tick Saliva"

_pathogens, 2022, doi:10.3390/pathogens11050530_

Round 1
Reviewer 1 Report
The chemical components in tick saliva responsible for Borrelia migration and uptake by the tick is an important topic of study, both for researchers interested in vector-pathogen interactions but also potentially for diagnosis of residual infection. I appreciated the three different methods tested and the inclusion of an appendix with detailed methods. I have no real problems with the science, but the manuscript was hard to follow due to journal formatting- for instance, the tables breaking across pages.
1.Table 1 could be reduced in size or eliminated altogether as no migration was observed in any condition.
2.In Table 2, it is unclear why some rows are shaded. I appreciate the caveats in the discussion (use of adult female saliva).
3.In Materials ans Methods, there are numerous place where B. burgdorferi is not italicized, and where a superscript is not used when discussing cell numbers (e.g., line 283, 2-3 x 10^7/ml).
4.Line 328, references should be Shih et al.
Author Response
Response to Reviewers
Thank you for your helpful review. These are all good points, to which we have responded accordingly. All responses are shown in italics.
The chemical components in tick saliva responsible for Borrelia migration and uptake by the tick is an important topic of study, both for researchers interested in vector-pathogen interactions but also potentially for diagnosis of residual infection. I appreciated the three different methods tested and the inclusion of an appendix with detailed methods. I have no real problems with the science, but the manuscript was hard to follow due to journal formatting- for instance, the tables breaking across pages.
Table 2 was reformatted to reduce the row height so that it can fit on one page.
1.Table 1 could be reduced in size or eliminated altogether as no migration was observed in any condition.
Table 1 does show all negative results. The point of including it is that we were unable to reproduce published findings. We have now included it as a Supplemental table so still accessible.
2.In Table 2, it is unclear why some rows are shaded. I appreciate the caveats in the discussion (use of adult female saliva).
Table 2 (now Table 1) is only shaded where tick saliva was tested; this is now included in the text. For Table 2 (former Table 3), the shading was completely removed to avoid confusion.
3.In Materials ans Methods, there are numerous place where B. burgdorferi is not italicized, and where a superscript is not used when discussing cell numbers (e.g., line 283, 2-3 x 10^7/ml).
These have been corrected
4.Line 328, references should be Shih et al.
There are actually 2 different references--Shih and Shi (#16 and 17)
Reviewer 2 Report
The manuscript (pathogens-1692123) entitled, “Borrelia burgdorferi Migration Assays for Evaluation of Chemo-attractants in Tick Saliva” by Jacobs from Monica Embers laboratory has conducted three variation of chemotactic response experiment with both in vitro grown and host-adapted Lyme spirochetes. It may have benefit for the xenodiagnoses if applied carefully. Overall, the manuscript is informative; however, some questions remain that need to be clarified. Furthermore, addressing the following comments will improve the usefulness of this article for other researchers.
1.The failure of U-tube assay could be because:
a. B. burgdorferi seem to be in rich BSKII culture medium. Why will spirochetes move to less nutritionally favorable environment?
b. Suspension of culture in PBS/2%BSA and gradient formation in the U-tube overnight before adding culture could have allowed spirochetes to move towards nutrients on the other side of U-tube. These could be the differences from other two assays where spirochetes may encounter nutrients gradient from low nutrient suspension buffer.
2. What is the difference between rows, 7, 9 and 12 in Table 3? This needs to be explained in text.
3. Line 115-116 vs. line 131-132. These results with respect to host-adapted Borrelia seem contradictory. Caveat should be added.
4. Lines 294-296. This does not seem accurate. Values mentioned here are too high. Authors describe the medium here to be reduced carbohydrate medium. Where did authors get these values? Reference should be provided. Standard BSKII medium contains 0.5% Glucose, 0.08% Sodium pyruvate and 0.04% N-acetyl-D-glucosamine.
Minor points:
- B. burgdorferi and Ixodes scapularis should be italicized throughout.
- Spirochete in the text often is mentioned as x107/ml. It should be corrected throughout to either 107 or 10^7.
- Line 283. Do the authors’ mean, ‘To one side----’ instead of ‘to either side---? Did they inoculate on both sides of U-tube?
- What is [CA]? I could not find its definition in the manuscript.
Author Response
We thank the reviewer for the thoughtful comments on our paper and have made revisions consistent with the reviewer concerns. All responses to the comments are shown in italics.
The manuscript (pathogens-1692123) entitled, “Borrelia burgdorferi Migration Assays for Evaluation of Chemo-attractants in Tick Saliva” by Jacobs from Monica Embers laboratory has conducted three variation of chemotactic response experiment with both in vitro grown and host-adapted Lyme spirochetes. It may have benefit for the xenodiagnoses if applied carefully. Overall, the manuscript is informative; however, some questions remain that need to be clarified. Furthermore, addressing the following comments will improve the usefulness of this article for other researchers.
1.The failure of U-tube assay could be because:
a. B. burgdorferi seem to be in rich BSKII culture medium. Why will spirochetes move to less nutritionally favorable environment?
b. Suspension of culture in PBS/2%BSA and gradient formation in the U-tube overnight before adding culture could have allowed spirochetes to move towards nutrients on the other side of U-tube. These could be the differences from other two assays where spirochetes may encounter nutrients gradient from low nutrient suspension buffer.
We agree, and this is why we tried to reduce the carbohydrate content and utilized PBS for the assays. We did not try overnight incubation beforehand, but the spirochetes were given ample time to migrate from the PBS to the sugar molecule chemoattractants on the other side of the u-tube with incubation times between 19-72 hours.
2. What is the difference between rows, 7, 9 and 12 in Table 3? This needs to be explained in text.
This Table has been revised to exclude shading so as to avoid confusion.
3. Line 115-116 vs. line 131-132. These results with respect to host-adapted Borrelia seem contradictory. Caveat should be added.
Both sentences indicate that tick saliva was a chemoattractant for host-adapted cells, but not for in vitro-grown Bb. if this is not clear, we can revise again.
4. Lines 294-296. This does not seem accurate. Values mentioned here are too high. Authors describe the medium here to be reduced carbohydrate medium. Where did authors get these values? Reference should be provided. Standard BSKII medium contains 0.5% Glucose, 0.08% Sodium pyruvate and 0.04% N-acetyl-D-glucosamine.
Thank you for pointing out the confusing wording. this has been revised to read : "a second variation included use of a modified BSK-II base medium such that the carbohydrates were reduced from the original concentrations to 50% of the standard amount of pyruvic acid, 10% of the glucose concentration and 50% of the N-acetyl D-glucosamine concentration. "
Minor points:
- B. burgdorferi and Ixodes scapularis should be italicized throughout.
- Spirochete in the text often is mentioned as x107/ml. It should be corrected throughout to either 107 or 10^7.
These have both been corrected
- Line 283. Do the authors’ mean, ‘To one side----’ instead of ‘to either side---? Did they inoculate on both sides of U-tube?
Good catch-thank you. We meant one side and this has been corrected.
- What is [CA]? I could not find its definition in the manuscript.
CA= chemoattractant. this is now a footnote in Table 1